# Discriminating Reflux from Non-Reflux Diseases of Superficial Veins in Legs by Novel Non-Contrast MR with QFlow Technique

**DOI:** 10.3390/jpm11040242

**Published:** 2021-03-26

**Authors:** Yuan-Hsi Tseng, Chien-Wei Chen, Min Yi Wong, Teng-Yao Yang, Bor-Shyh Lin, Hua Ting, Yao-Kuang Huang

**Affiliations:** 1Division of Thoracic and Cardiovascular Surgery, Chiayi Chang Gung Memorial Hospital, College of Medicine, Chia-Yi and Chang Gung University, Taoyuan 33302, Taiwan; 8802003@cgmh.org.tw (Y.-H.T.); mynyy001@gmail.com (M.Y.W.); 2Department of Diagnostic Radiology, Chiayi Chang Gung Memorial Hospital, College of Medicine, Chia-Yi and Chang Gung University, Taoyuan 33302, Taiwan; chienwei33@gmail.com; 3Institute of Medicine, Chung Shan Medical University, Taichung 408, Taiwan; huating@csmu.edu.tw; 4Institute of Imaging and Biomedical Photonics, National Chiao Tung University, Tainan 300, Taiwan; borshyhlin@gmail.com; 5Department of Cardiology, Chiayi Chang Gung Memorial Hospital, College of Medicine, Chia-Yi and Chang Gung University, Taoyuan 33302, Taiwan; 2859@adm.cgmh.org.tw

**Keywords:** MRI, non-contrast, venography, TRANCE, QFlow

## Abstract

**Objectives**: To find an objective diagnostic tool for the superficial veins in legs. **Methods**: This study included 137 patients who underwent TRANCE-MRI from 2017 to 2020 (IRB: 202001570B0). Among them, 53 with unilateral leg venous diseases underwent a QFlow scan and were classified into the reflux and non-reflux groups according to the status of the great saphenous veins. **Results**: The QFlow, namely stroke volume (SV), forward flow volume (FFV), mean flux (MF), stroke distance (SD), and mean velocity (MV) measured in the external iliac, femoral, popliteal, and great saphenous vein (GSV). The SV, FFV, SD, MF, SD, and MV in the GSV (morbid/non-morbid limbs) demonstrated a favorable ability to discriminate reflux from non-reflux in the ROC curve. The SD in the GSV and GSV/PV ratio (*p* = 0.049 and 0.047/cutoff = 86 and 117.1) and the MV in the EIV/FV ratio, GSV, and GSV/PV ratio (*p* = 0.035, 0.034, and 0.025/cutoff = 100.9, 86.1, and 122.9) exhibited the ability to discriminate between reflux and non-reflux group. The SD, MV, and FFV have better ability to discriminate a reflux from non-reflux group than the SV and MF. **Conclusions**: QFlow may be used to verify the reflux of superficial veins in the legs. An increasing GSV/PV ratio is a hallmark of reflux of superficial veins in the legs.

## 1. Background

Patients were suspected as having a disease of venous origin when they came with tortuous varicose veins on calves, asymmetric swollen legs, and watery ulcers around the gaiter area of their feet. Venous diseases in those legs may be caused by superficial venous reflux from valvular dysfunction, occlusion of the deep venous system by thrombi, and compression of the pelvic mass. The resulting signs include spider veins, foot pigmentation, claudication, stasis ulcers, swollen limbs with deep vein thrombosis, and the fatal consequence of the pulmonary emboli [1,2,3,4,5,6]. Ultrasound examination (US) is the standard procedure for venous disease of the legs and could provide most information by experienced operators [7,8]. Few tools are available for objective venous evaluation in lower limbs. The venous system is not exactly enhanced by computed tomography (CT) venography, and high-quality enhancement requires specific access from the morbid limb. Compared with conventional angiography, most magnetic resonance venography (MRV) techniques that involve contrast media have been proven to be more sensitive towards the detection of lesions in vessels [9]. Triggered angiography non-contrast-enhanced (TRANCE) magnetic resonance imaging (MRI) can record differences in vascular signal intensity during the cardiac cycle for subsequent image subtraction and obtain vascular images without the use of contrast media (Figure 1). All images of the arterial systems are reconstructed by three-dimensional turbo spin-echo (TSE) at systolic and diastolic periods. During systole, arterial blood flows rapidly, and the arteries are black. In the diastole phase, blood flow in the arteries is slow, and the arteries are bright. Subtraction of the two-phased scans made up a 3D data set with only arteries (MRA). Another image of the venous systems (MRV) was evaluated by 3D TSE short tau inversion recovery (STIR) during the systolic period. STIR provides additional background suppression because the fat and bones are also suppressed. This technique was initially evaluated for pelvic vessels. Our clinical applications of this technique have innovated the anatomy of the whole venous system in legs since 2017 and further proven its efficiency in different venous scenarios thereafter [6,10,11]. TRANCE-MRI helps to not only exclude venous compression but also reveal the major tributaries, thus providing better guidance for venous ablation. In this study, we evaluated the possible correlation between reflux with ultrasound and morphologic changes from the TRANCE-MRI and studied the hemodynamic patterns obtained from QFlow analysis through TRANCE-MRI to help differentiate reflux in superficial veins further.

## 2. Methods

### 2.1. Patients

The study protocol was approved by the Institutional Review Board (IRB) of Chang Gung Memorial Hospital (IRB number: 202001570B0). The study recruited consecutive patients who underwent TRANCE-MRI for venous pathology in their legs at a tertiary hospital between April 2017 and September 2020. We analyzed their data to determine their clinical significance. All patients were suspected of having venous pathology in their legs. Patients were excluded if they exhibited poor compliance or had multiple comorbidities that made them unable to lie down for the 1 h TRANCE-MRI protocol. At first, 137 patients were enrolled for leg venous evaluation through TRANCE-MRI. Twelve patients were excluded for reasons including pregnancy, presence of non-MRI-compatible ferromagnetic implants, significant arrhythmia, and restless legs. The remaining 125 patients underwent anatomical evaluation through TRANCE-MRI. QFlow analysis and hemodynamic evaluation were performed in 53 patients with unilateral symptoms. These 53 patients were further categorized into the reflux and non-reflux groups according to the screening duplex and referral indication for TRANCE-MRI (Figure 2).

All 53 patients received noninvasive color Doppler ultrasonography (US) for the venous status in their lower legs before the scheduled TRANCE-MRI. The duplex examination was performed in the supine position, and the femoral vein, great saphenous vein (GSV), popliteal vein, and perforating vein in the calves were examined. Intra-abdominal and pelvic veins were not evaluated in all of the duplex examinations.

### 2.2. MRI Acquisition

MRI was performed using a 1.5 T MRI scanner (Philips Ingenia, Philips Healthcare, Best, The Netherlands) and a peripheral pulse unit trigger, with the patients in the supine position. All arterial system images were evaluated through a three-dimensional (3D) turbo spin-echo (TSE) technique during systole and diastole periods. TSE TRANCE imaging was executed using the following parameters: repetition time (TR), 1 beat; echo time (TE), shortest; flip angle, 90°; voxel size, 1.7 × 1.7 × 3 mm^3^; and field of view (FOV), 350 × 420. During systole, the arterial blood flow was relatively fast, which caused signal dephasing and lead to flow voids. Accordingly, when a systolic trigger was applied, the arteries would appear black. During diastole, the arterial blood flow was slow, and therefore, the signal was not dephased, and the arteries appeared bright on diastolic scans. Subtraction of the two-phased scans yielded a 3D data set of the arteries only. Other images of the venous systems were evaluated through 3D TSE short tau inversion recovery (STIR) during the systole period. TSE STIR TRANCE imaging was executed using the following parameters: TR, 1 beat; TE, 85; inversion recovery delay time, 160; voxel size, 1.7 × 1.7 × 4 mm^3^; and FOV, 360 × 320. STIR provides additional background suppression because the fat and bones are also suppressed. When a systolic trigger was applied, the arteries appear black. This imaging process yielded a 3D data set of the venous system, with no subtraction required. A quantitative flow scan was routinely performed to determine the appropriate trigger delay times for systolic and diastolic triggering. All images were acquired without the use of a gadolinium-based contrast medium. QFlow scans yielded multiple acquisitions within one cardiac cycle, resulting in multiple phases (Figure 3).

QFlow analysis included the following parameters: stroke volume (SV), forward and backward flow volumes, flux, stroke distance (SD), mean velocity (MV), and vessel area. In this study, a postprocessing package was used to calculate quantitative information such as flow velocity and allow the visualization as 2D flow maps overlaid on anatomic references. The external iliac veins, femoral veins, popliteal veins, and greater saphenous veins were analyzed.

### 2.3. Statistical Analysis

The continuous variables (age and QFlow) were analyzed using an unpaired two-tailed Student’s *t* test or one-way analysis of variance, and the discrete variables (sex, substance usage, comorbidities, and intervention history) were compared using a two-tailed Fisher’s exact test. The QFlow parameters (including SV, forward flow volume (FFV), mean flux (MF), SD, and MV in each venous segment) and the obstructive venous diseases were evaluated using receiver operating characteristic (ROC) curve analysis. All statistical analyses were conducted using the STATA statistics/Data Analysis (version 8.0; Stata Corporation, College Station, TX, USA). The results are presented as means and standard deviations. Statistical significance was defined as *p* < 0.05.

## 3. Results

The demographic and medical data of the 53 patients, including regarding sex, age, substance use, and comorbidities, are summarized in Table 1.

The mean age in the cohort was 59.7 ± 14.1 years, and the majority were women (38/53, 71.7%). The patients were further divided into the patients with reflux in the great saphenous vein (reflux group) and patients without reflux in the great saphenous vein (non-reflux group) according to their presenting symptoms and duplex findings (Figure 4).

The two groups were similar in terms of age, sex, substance use, comorbidities, and co-existence of leg ulcers. QFlow analysis through TRANCE-MRI involved the evaluation of SV (mL), FFV (mL), MF (mL), SD (cm), and MV (cm) in the external iliac veins, femoral veins, popliteal veins, and GSVs in the 53 patients. The superficial veins in the non-reflux group were mostly in normal size, with competent valve function. To minimize individual bias in the QFlow analysis, we compared the morbid and non-morbid limbs of the same patients. The performance of the QFlow parameters (SV, FFV, MV, SD, and MV) in the discrimination of the reflux from non-reflux patients was assessed using the ratio of the morbid limb to normal limb in each venous segment (Table 2).

The SV exhibited favorable discriminating ability for reflux in the GSV segment (AUC = 68.8%, 95% CI = 52.7–85%). The FFV had favorable discriminating ability for reflux in the GSV ratio (AUC = 74.9%, 95% CI = 58.2–86.5%) and GSV/PV ratio (AUC = 74%, 95% CI = 58.5–89.6%). The MF also effectively discriminated reflux from non-reflux in the GSV segment (AUC = 68.6%, 95% CI = 52.4–84.8%) and GSV/PV ratio (AUC = 66.7%, 95% CI = 50.4–83%). The SD demonstrated favorable discrimination performance for venous reflux in the EIV/FV ratio (AUC = 66.5%, 95% CI = 51.3–81.7%), GSV segment (AUC = 68.6%, 95% CI = 53.0–84.2%), and GSV/PV ratio (AUC = 66.8%, 95% CI = 53.0–84.7%). The MV demonstrated favorable discrimination performance for venous reflux in the EIV/FV ratio, (AUC = 67.8%, 95% CI = 52.5–83.1%), GSV segment, (AUC = 70%, 95% CI = 54.6–85.4%), and GSV/PV ratio (AUC = 71.2%, 95% CI = 55.7–86.7%). The SV exhibited a significant ability to discriminate between reflux and non-reflux venous diseases in the GSV segment (*p* = 0.047). The FFV in the GSV segment and GSV/PV ratio exhibited a significant ability to discriminate between reflux and non-reflux venous diseases (*p* = 0.008 and 0.011, respectively).

The MF in the GSV segment exhibited a significant ability to discriminate between reflux and non-reflux venous diseases (*p* = 0.049). The SD in the GSV and GSV/PV ratio exhibited a significant ability to discriminate between reflux and non-reflux venous diseases (*p* = 0.049 and 0.047, respectively, and cutoff = 86 and 117.1, respectively). Furthermore, the MV in the EIV/FV ratio, GSV segment, and GSV/PV ratio exhibited a significant ability to discriminate between reflux and non-reflux venous diseases (*p* = 0.035, 0.034, and 0.025, respectively, and cutoff = 100.9, 86.1, and 122.9, respectively). The SD, MV, and FFV have better ability to discriminate reflux from non-reflux venous diseases than the SV and MF by the QFlow analysis. The ROC curve of SD and MF is shown in Figure 5.

## 4. Discussion

Most patients with suspected venous diseases of the legs undergo air plethysmography (APG) and ultrasound examination (US) at the beginning of therapy. Air plethysmography is a non-invasive tool for quantifying venous reflux and obstruction by measuring volume changes in the leg. US, a rapid tool, could also provide additional information on active reflux and gravitational reflux through standing position by experienced operators. US is operator-dependent and lacking information of pelvic and abdominal areas. However, in many institutions, including ours, the duplex is performed exclusively in ultrasound centers and not done by the same physician in the clinic, which requires additional communication between staff to gain sufficient information.

Intravenous ultrasonography (IVUS) is a new imaging modality for diagnosing deep vein disease and is mostly guiding effective endovascular treatment in iliac and caval venous obstructive lesions [12,13]. However, IVUS is invasive and only provides details of the venous lumen without information of the superficial venous system. Venography was considered as standard for the detection of deep venous thrombosis and other venous occlusive diseases, but venography cannot display the superficial veins outside the drainage course of the contrast-media injection site. CT venography may be useful for the exclusion of pulmonary embolism in patients with signs of deep venous thrombosis in the legs; however, CT venography also requires the injection of contrast media in the morbid limb to achieve optimal venous imaging of the extremities, which can be dangerous for the diseased limb [14]. Magnetic resonance angiography (MRA) techniques for reconstructing vascular structures include time-of-flight (TOF), phase-contrast, and ECG-gated TSE MRA. The major disadvantage of TOF-MRV is that the FOV is small for each image obtained and that it requires considerable time to image the whole lower extremity. MRI with gadolinium-based contrast media is a relatively rapid method for imaging of the lower extremities [15,16]. Although MRI does not involve radiation exposure, the noniodinated contrast agents involved in the process have undesirable effects on the patients. For instance, nephrogenic systemic fibrosis is a dangerous condition caused by exposure to gadolinium-based contrast agents in patients with pre-existing impairment of kidney function and even in patients with normal renal function [17,18]. Phase-contrast MRI (PC-MRI) depends on phase shifts caused by blood flow. Thus, this technique permits the use of coronal or sagittal slice orientations with an FOV along the direction of the vessel of interest and can quantitatively measure the flow dynamics of the region of interest. Most studies have applied PC-MRA for evaluating central nervous system pathology including the hydrocephalus [19,20].

Imaging vascular structures of the whole lower extremity using traditional MRA techniques, such as TOF-MRA and PC-MRA, is time-consuming. The ECG-gated, multistep TSE technique (i.e., TRANCE-MRI) offers the possibility of imaging vascular structures of the whole lower extremity in a relatively short time in clinical settings. For ECG-gating, different imaging times are required for different flow characteristics, and therefore, the image quality can be optimized faster. Most related studies on non-contrast-enhanced MR have used this technique to evaluate arterial diseases [21,22,23,24,25]. Our team has innovated the use of TRANCE-MRI for obtaining more valuable information for the management of complicated lower venous diseases [4,5,6,11]. Moreover, experienced radiological teams may require an examination time of less than 30 min. Thus, 3D imaging of the morphology of the venous anatomy of the lower extremities, particularly low-flow superficial venous systems, is possible without the use of contrast media or radiation. TRANCE-MRI is not very expensive in our healthcare system (less than 250 USD/each examination) and has become one of the regular survey methods for venous diseases in our institution today (Appendix A).

In contrast to a CT angiogram, TRANCE-MRI could be used to perform hemodynamic estimation. In this study, we further analyzed the QFlow data of the segmental leg veins. QFlow analysis through TRANCE-MRI involved the evaluation of SV (mL), FFV (mL), MF (mL), SD (cm), and MV (cm) in the external iliac veins, femoral veins, popliteal veins, and GSVs in the 53 patients with unilateral leg symptoms. Those results revealed that the MV and SD were more sensitive than SV and MF to the differentiation of reflux from non-reflux venous diseases in superficial veins.

QFlow hemodynamic patterns differ between reflux and non-reflux venous diseases mainly in terms of the GSV segment and GSV/PV ratio. In other words, a higher GSV/PV ratio (i.e., GSV/PV > 1 in TRANCE-MRI) is highly correlated to reflux in GSV. The differences were noted in all QFlow parameters, including SV, FFV, MF, SD, and MV. The MF in the GSV segment exhibited a significant ability to discriminate between reflux and non-reflux venous diseases (*p* = 0.049). The SD in the GSV segment and GSV/PV ratio exhibited a significant ability to discriminate between reflux and non-reflux venous diseases (*p* = 0.049 and 0.047, respectively, and cutoff = 86 and 117.1, respectively). The MV in the EIV/FV ratio, GSV segment, and GSV/PV ratio demonstrated a significant ability to discriminate between reflux and non-reflux venous diseases (*p* = 0.035, 0.034, and 0.025, respectively, and cutoff = 100.9, 86.1, and 122.9, respectively). Notably, the non-reflux venous group exhibited a higher EIV/FV ratio for MV. Obstructive venous diseases such as May–Thurner syndrome, external compression, and venous thrombosis account for at least part of the non-reflux venous leg diseases.

### Study Limitations

The major limitations of this study are its non-randomized design and small sample size. Second, only patients with typical, unilateral leg disease were included. Third, the TRANCE-MR and US were performed in the supine position, without details of gravitational and muscular interactions. Retrograde flow is possibly not provoked during the TRANCE-MR examination and could be better reproduced by US in a standing position. However, this is the largest series to discuss the use of TRANCE-MRI as an imaging tool for venous diseases of the lower extremities. In addition to proving the morphological advantage and safety of TRANCE-MRI, this study is the first attempt to analyze QFlow in the clinical scenarios of the superficial venous reflux of the lower extremities.

## 5. Conclusions

TRANCE-MRI is a potential tool for verifying the characteristic ultrasound features of reflux venous diseases in the lower extremities. A higher GSV/PV ratio could be a hallmark in the QFlow study of reflux venous diseases in the legs through TRANCE-MRI.

## 6. Patents

This project is under the reviewing process in the Taiwan Intellectual property Office. (No 109126307).

## Figures and Tables

**Figure 1 jpm-11-00242-f001:**
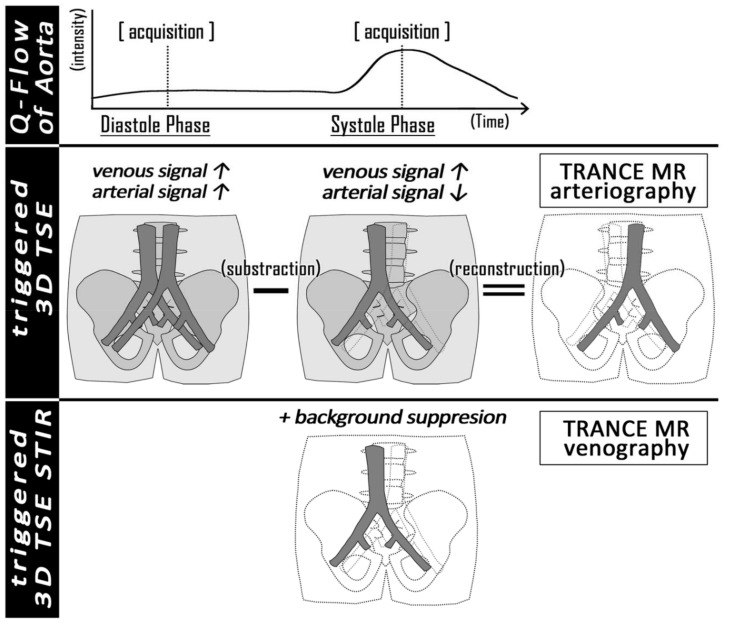
The principle of TRANCE-MRI: We use a peripheral pulse unit to identify the systolic and diastolic phases (**upper row**). The diastolic phase subtracting the systolic phase simulated arteriography (**middle row**). The systolic 3D TSE was further modified by background suppression (bone, soft tissue, and fat) to gain TRANCE-MR venography.

**Figure 2 jpm-11-00242-f002:**
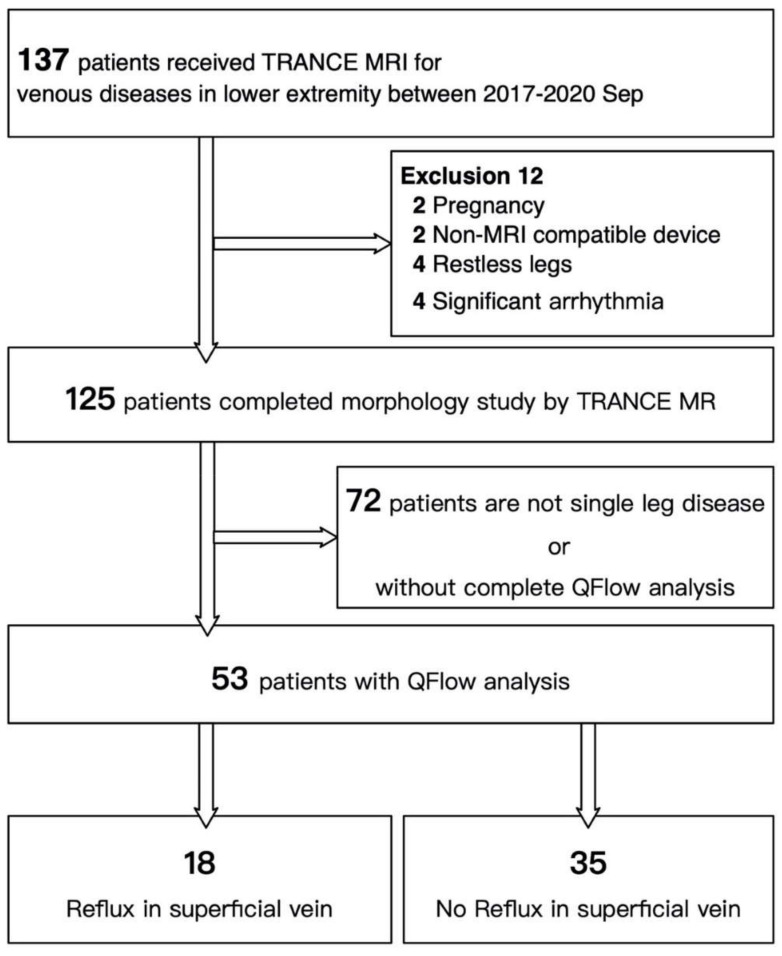
Flowchart of cohort selection.

**Figure 3 jpm-11-00242-f003:**
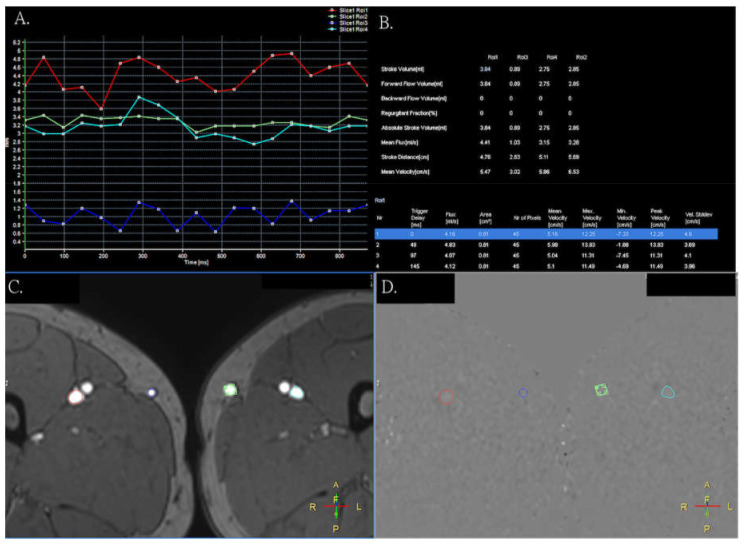
QFlow scanning through TRANCE-MRI including stroke volume (SV), forward flow volume (FFV), mean flux (MF), stroke distance (SD), and mean velocity (MV). (**A**) Two great saphenous veins and two popliteal veins with the flow sequence by time. (**B**) QFlow parameters with different trigger delays. (**C**) Areas of interest in both great saphenous and popliteal veins (blue: right side, normal GSV; green: left side, diseased GSV). (**D**) Image obtained during data acquisition.

**Figure 4 jpm-11-00242-f004:**
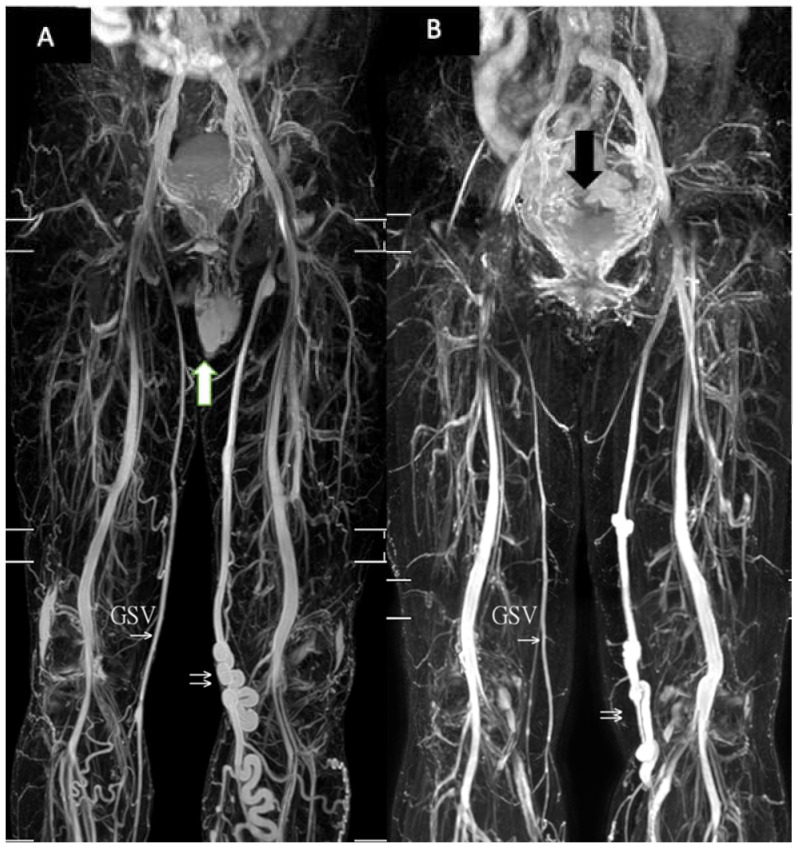
Typical patients with reflux in the great saphenous vein by TRANCE-MR. (**A**) A 56-year-old man with reflux of the left great saphenous vein. The white arrow indicates the testis. GSV: normal right great saphenous vein. The double arrow indicates the diseased great saphenous vein with its tributaries. (**B**) A 67-year-old woman also with reflux in the left great saphenous vein. The black arrow indicates the uterine. GSV: normal right side great saphenous vein. The double arrow indicates the diseased great saphenous vein with its tributaries.

**Figure 5 jpm-11-00242-f005:**
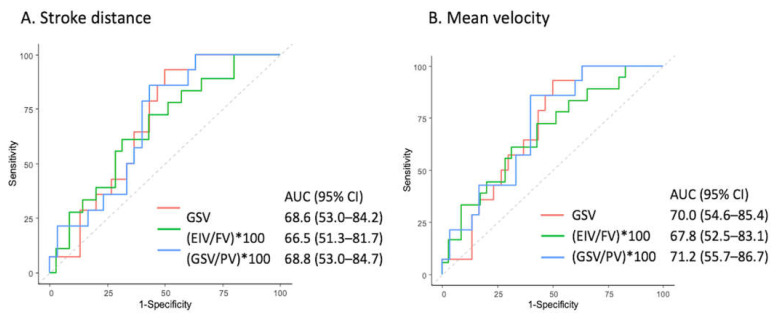
ROC examination of the QFlow parameters of (**A**) stroke distance and (**B**) mean velocity.

**Table 1 jpm-11-00242-t001:** Clinical characteristics of the patients who underwent QFlow analysis by TRANCE-MRI by the presence of reflux in superficial veins.

Variable	Total (*n* = 53)	Reflux (*n* = 18)	No Reflux (*n* = 35)	*p* Value
Male sex	15 (28.3)	7 (38.9)	8 (22.9)	0.334
Age, year	59.7 ± 14.1	56.9 ± 12.9	61.1 ± 14.7	0.317
Substance use				
Smoking	10 (18.9)	5 (27.8)	5 (14.3)	0.279
Alcohol	3 (5.7)	1 (5.6)	2 (5.7)	1.000
Betel nuts	3 (5.7)	2 (11.1)	1 (2.9)	0.263
Comorbidities				
Hypertension	15 (28.3)	3 (16.7)	12 (34.3)	0.215
Diabetes	8 (15.1)	2 (11.1)	6 (17.1)	0.701
Coronary artery disease	1 (1.9)	0 (0.0)	1 (2.9)	1.000
Deep vein thrombosis	6 (11.3)	0 (0.0)	6 (17.1)	0.085
Cancer	8 (15.1)	1 (5.6)	7 (20.0)	0.240
Left leg involved	32 (60.4)	8 (44.4)	24 (68.6)	0.138
Chronic leg ulcer	8 (15.1)	5 (27.8)	3 (8.6)	0.104

Data are presented as a percentage or mean ± standard deviation.

**Table 2 jpm-11-00242-t002:** The performance of QFlow parameters (the ratio of the morbid limb to normal limb) in discriminating reflux in superficial veins of legs.

Variable	Total (*N* = 53)	Reflux (*n* = 18)	No Reflux (*n* = 35)	*p*	AUC, % (95% CI) *	Cutoff #	Sensitivity, % (95% CI)	Specificity, % (95% CI)
SV								
EIV	100 (60, 124)	102 (78, 119)	82 (49, 132)	0.367	55.0 (37.3–72.7)	NA	NA	NA
FV	91 (60, 136)	101 (64, 165)	80 (52, 128)	0.237	57.1 (38.6–75.7)	NA	NA	NA
(EIV/FV) * 100	86 (58, 147)	99 (66, 147)	73 (53, 157)	0.419	60.7 (42.9–78.5)	NA	NA	NA
GSV	130 (72, 766)	276 (126, 875)	105 (59, 600)	0.047 *	68.8 (52.7–85.0) *	>108.6	85.7 (57.2–98.2)	53.3 (34.3–71.7)
PV	89 (53, 120)	78 (51, 207)	92 (53, 120)	0.851	52.9 (32.5–73.2)	NA	NA	NA
(GSV/PV) * 100	154 (79, 518)	257 (152, 563)	126 (61, 472)	0.087	66.2 (49.7–82.7)	NA	NA	NA
FFV								
EIV	92 (56, 121)	101 (70, 116)	84 (47, 132)	0.430	53.8 (36.2–71.4)	NA	NA	NA
FV	95 (64, 136)	101 (64, 165)	92 (58, 128)	0.320	56.7 (38.1–75.3)	NA	NA	NA
(EIV/FV) * 100	81 (49, 145)	99 (61, 147)	72 (45, 124)	0.348	58.6 (41.0–76.1)	NA	NA	NA
GSV	125 (81, 332)	276 (126, 821)	108 (64, 183)	0.008 *	74.9 (59.3–90.4) *	>114.3	85.7 (57.2–98.2)	60 (40.6–77.3)
PV	92 (63, 120)	78 (52, 207)	97 (71, 120)	0.693	55.0 (34.5–75.5)	NA	NA	NA
(GSV/PV) * 100	147 (73, 312)	260 (156, 563)	109 (63, 222)	0.011 *	74.0 (58.5–89.6) *	>222.0	71.4 (41.9–91.6)	76.7 (57.7–90.1)
MF								
EIV	100 (59, 124)	102 (70, 119)	82 (49, 133)	0.419	55.2 (37.6–72.8)	NA	NA	NA
FV	95 (64, 136)	102 (64, 164)	89 (59, 127)	0.302	57.6 (39.1–76.1)	NA	NA	NA
(EIV/FV) * 100	83 (55, 146)	96 (65, 147)	71 (53, 124)	0.338	61.2 (43.4–79.0)	NA	NA	NA
GSV	129 (67, 778)	276 (125, 847)	106 (60, 600)	0.049 *	68.6 (52.4–84.8) *	>109.3	85.7 (57.2–98.2)	53.3 (34.3–71.7)
PV	88 (61, 120)	77 (51, 209)	92 (65, 120)	0.778	53.8 (33.4–74.2)	NA	NA	NA
(GSV/PV) * 100	160 (80, 538)	257 (157, 569)	119 (61, 507)	0.078	66.7 (50.4–83.0) *	>136.5	78.6 (49.2–95.3)	60.0 (40.6–77.3)
SD								
EIV	98 (71, 123)	101 (74, 123)	88 (57, 128)	0.237	61.0 (43.0–78.9)	NA	NA	NA
FV	94 (76, 128)	94 (76, 128)	94 (70, 129)	0.910	53.8 (35.6–72.0)	NA	NA	NA
(EIV/FV) * 100	94 (64, 122)	106 (77, 163)	82 (51, 119)	0.051	66.5 (51.3–81.7) *	>101.0	61.1 (35.7–82.7)	68.6 (50.7–83.1)
GSV	126 (73, 423)	224 (118, 765)	93 (50, 365)	0.049 *	68.6 (53.0–84.2) *	>86	92.9 (66.1–99.8)	50.0 (31.3–68.7)
PV	91 (57, 119)	74 (53, 111)	98 (64, 120)	0.276	59.8 (38.8–80.7)	NA	NA	NA
(GSV/PV) * 100	135 (66, 333)	204 (133, 582)	112 (39, 318)	0.047 *	68.8 (53.0–84.7) *	>117.1	85.7 (57.2–98.2)	56.7 (37.4–74.5)
MV								
EIV	99 (71, 128)	101 (73, 151)	88 (57, 128)	0.176	61.0 (43.0–78.9)	NA	NA	NA
FV	94 (76, 128)	94 (76, 128)	94 (70, 129)	0.910	53.8 (35.6–72.0)	NA	NA	NA
(EIV/FV) * 100	94 (63, 129)	109 (77, 165)	82 (51, 119)	0.035 *	67.8 (52.5–83.1) *	>100.9	61.1 (35.7–82.7)	68.6 (50.7–83.1)
GSV	126 (73, 382)	300 (118, 768)	93 (56, 338)	0.034 *	70.0 (54.6–85.4) *	>86.1	92.3 (66.1–99.8)	50.0 (31.3–68.7)
PV	91 (57, 119)	74 (54, 111)	98 (64, 120)	0.276	59.8 (38.8–80.7)	NA	NA	NA
(GSV/PV) * 100	135 (66, 332)	206 (133, 585)	112 (40, 318)	0.025 *	71.2 (55.7–86.7) *	>122.9	85.7 (57.2–98.2)	60.0 (40.6–77.3)

Data are presented as median [25th percentile, 75th percentiles]; AUC, area under curve; CI, confidence interval; FFV, forward flow volume; MF, mean flux; MV, mean velocity; SD, stroke distance; SV, stroke volume; NA, not applicable; # Determined by the Youden index; * *p* value < 0.05.

## Data Availability

The data presented in this study are available on request from the corresponding author. The data are not publicly available due to ethical restrictions.

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
