# Peer review of "Discriminating Reflux from Non-Reflux Diseases of Superficial Veins in Legs by Novel Non-Contrast MR with QFlow Technique"

_jpm, 2021, doi:10.3390/jpm11040242_

Round 1

Reviewer 1 Report

Even though Duplex-ultrasound is operator-dependent, experienced examiners can perform very accurate and detailed venous mapping of the lower extremities. Ultrasound is readily available in the examination room and has no side-effects what so ever for the patients. It can also be done in less then 30 minutes (the time described by the authors for the MR-investigation) an it provides a real-time picture of the venous system showing hemodynamic effects (for example reflux) in a dynamic fashion. 

It is therefore not really conceivable why someone would utilize a far more expensive and logistically challenging examination. It might be interesting for scientific purposes, but it doesn't seem to be applicable in day to day practice 

This should be discussed.

Author Response

Dear Editors, Reviewers and

We are submitting our manuscript entitled “Discriminating superficial venous valvular insufficiency from non-insufficiency diseases by novel Non-contrast MR with QFlow technique " for consideration of “Journal of Personalized Medicine” after revise.

        Thank you very much again for granting the privilege to us to revise the paper. We have specifically responded to the reviewers’ questions and criticisms point-by-point as follows and add them into this version. Any changes in the manuscript can be tracked by the tool of the MS Word and be marked by underline.

Reviewer 1

[Comment 1]

Even though Duplex-ultrasound is operator-dependent, experienced examiners can perform very accurate and detailed venous mapping of the lower extremities. Ultrasound is readily available in the examination room and has no side-effects what so ever for the patients. It can also be done in less than 30 minutes (the time described by the authors for the MR-investigation) and it provides a real-time picture of the venous system showing hemodynamic effects (for example reflux) in a dynamic fashion.

[Answer 1]

Thank for your thoughtful comment. We totally agree that an Ultrasound though an experienced operator will be more informative, especially in the gravity respond on the standing position. However, the ultrasound machine is not available in the surgical outpatient department in our institution. We add these opinions in the “discussion section”

[Change]

US, a rapid tool, could also provide additional information of active reflux and gravitational reflux though standing position by experienced operators. However, US is operator-dependent, time-consuming, and lacking information of pelvic and abdominal areas.

[Comment 2]

It is therefore not really conceivable why someone would utilize a far more expensive and logistically challenging examination. It might be interesting for scientific purposes, but it doesn't seem to be applicable in day to day practice

[Answer 2]

TRANCE MR did not require contrast media, thus is not very expansive in our health care systems (less than 200 USD/each exam, mostly cover by national insurance system). Meanwhile, this exam enhances the communication between doctors and those patients and became our regular survey for venous diseases now. We had added these descriptions into discussion.

[Change]

The TRANCE-MRI is not very expansive in our health care system (less than 250 USD/each examination) and become one of regular survey for venous diseases in our institution today.

the complete response to the reviewers, please see attach files.

Reviewer 2 Report

Comments on

General:

Very interesting topic, as it is, indeed, difficult to evaluate venous disease “Obectively”. And TRANCE MRI seems to be an interesting point, the images are gorgeous.

The authors mix superficial and deep venous disease. The whole article should be rewritten separating both.

Abstract:

methods: “Limb disease” please specify: Superficial reflux? Deep obstruction or reflux?

Background:

The title of your article refers to “Superficial Veins” – why then do you only measure in GSV (one superficial) but not in SSV, not in tributaries? Instead in three deep veins, one from the pelvic system? Of course, the deep veins are interesting when studying superficial ones. But then you only study one?

You talk about reflux and obstruction very indistinctly, but obstruction is a primary deep vein problem (seldom also superficial), whereas Reflux is THE problem with superficial venous insufficiency.

When you talk about different ways to measure venous insufficiency and obstruction you cannot leave out Duplex Ultrasound or Air Plethysmography, which are the most used (the first) and the only one exactly repeatable and reliable until now (the second).

For sure duplex Ultrasound it is not „objective“, as it is operator dependent, but this, at least has to be mentioned. And your new method, even if the images are gorgeous, aren’t they also operator dependent as far as the interpretation is “human”? Thus, you cannot simply leave out the most used tool in your list.

TRANCE-MRI helps to not only locate the venous compression but also reveal the major collateral veins, thus providing better guidance for venous ablation.

In this sentence you mix topics: Compression and collateral veins are the domain of deep venous obstruction – and these deep veins NEVER become ablated!

It is in the superficial veins, where we perform ablation (or multiple other treatment options you seem not to be familiar with) – but superficial veins never get incompetent due to Compression …?? It looks like you are coming from deep vein diagnostics and apply the tool to superficial veins, which you seem not to be familiar with. This is a pity, because I am sure, that what you present is of major interest.

Then you talk about valvular and non-valvular insufficiency… where might I instruct myself in non-valvular insufficiency of superficial leg veins? (I assume we are talking about superficial veins, as this is the name of your article…?

If this is something new, you want to introduce to the reader, you have to describe it in the background.

To summarize: Please rewrite the Background with the help of somebody treating superficial veins since years and introduce all what is needed to understand the rest.

I cannot find methods “in the typical location” after background and before Results – I have been reading and re-reading results without understanding anything, just to find methods later on… Methods is always prior to results!

You write that the patients had a Duplex in supine position prior to MRI – Please explain, why you do this non according to World-Wide Consensus from 2006!! (Coleridge Smith et al) which say, that superficial leg veins have to be explored in standing position?

The MRI investigation is performed in recumbent persons too, please describe, how you provoke retrograde flow? If not in any standardized way, the complete measurement lack reproducibility! Venous reflux occurs in standing position and you can apply different methods. Calf compression with standardized cuffs are the only one repeatable with always the same results as to my knowledge.

Results

Image 1 a and b

For me it is obvious, which one is the GSV in the legs. Could you please add arrows to the findings – the healthy one on the right leg (both), the healthy one below the onset of reflux in tributaries at knee level left and the reflux in tributaries?

In table 2 I get lost – suddenly you talk about obstructive and non obstructive – but if you are talking about superficial leg vein reflux discrimination, for the first step you should have excluded those with deep vein obstruction. Of course, I know, that some have both – but if you introduce a new tool (and it looks like a good one), you should first publish the tool in “simple” Legs. As I do not know the methods, I cannot interprete the findings. Then you always talk about non valvular incompetence, which hasn’t been introduced to the reader and I personally do not know even though performing Phlebology since 25 years…

Then, I do not understand the table – you wanted to compare healthy to not healthy, but I only find data of insufficiency, where is the comparison to contralateral?

In the Flow chart you write “Non insufficiency” – but I thought, you include only people with insufficiency at one leg??

Discussion

You write, Duplex is time consuming (in experienced hands 10 minutes the most to map a patient…?) – but this investigation takes !! one hour!! (so your description in methods: You excluded persons that could not lay down for one hour). You describe IVUS – which has nothing to do with superficial vein disease.

Please sort out all those indications concerning only deep vein disease.

Even though I have tried to read and re-read the discussion, it should be rewritten leaving out all the deep venous problems, you continuously mix deep and superficial commenting on GSV reflux in the same sentence with May-Turner (which should be subject of another article).

In your Text Conclusion you do not write about consequences, in the Abstract yes (see comment above).

Author Response

Reviewer 2

[Comment 1]

Very interesting topic, as it is, indeed, difficult to evaluate venous disease “Objectively”. And TRANCE MRI seems to be an interesting point, the images are gorgeous.

The authors mix superficial and deep venous disease. The whole article should be rewritten separating both.

[Answer 1]

We appreciated your informative comments for this article. We defined the group in this version, to reduce the misleading.

Change:

The patients were further divided into the superficial venous valvular insufficiency (valvular insufficiency group)and without superficial venous valvular insufficiency (non-valvular insufficiency groups) according to their presenting symptoms and Duplex finding.

[Comment 2]

Abstract:

methods: “Limb disease” please specify: Superficial reflux? Deep obstruction or reflux?

[Answer 2]

We revised the paragraph as your suggestion.

Change: Abstract, Methods, line 2

53 with unilateral superficial venous reflux in the leg.

[Comment 3]

Background:

The title of your article refers to “Superficial Veins” – why then do you only measure in GSV (one superficial) but not in SSV, not in tributaries? Instead in three deep veins, one from the pelvic system? Of course, the deep veins are interesting when studying superficial ones. But then you only study one?

[Answer 3]

The major reason is that the GSVs are more constant than SSVs in supine position by 1.5T MRI. We just started to analyze SSV recently but did not gain meaningful conclusion yet.

[Comment 4]

You talk about reflux and obstruction very indistinctly, but obstruction is a primary deep vein problem (seldom also superficial), whereas Reflux is THE problem with superficial venous insufficiency.

[Answer 4]

Thank for your comments.

We specified the reflux as “superficial venous reflux” in this version, to reduce misunderstanding of the readers.

Changed: Conclusion line 262 to 265

[Comment 5]

When you talk about different ways to measure venous insufficiency and obstruction you cannot leave out Duplex Ultrasound or Air Plethysmography, which are the most used (the first) and the only one exactly repeatable and reliable until now (the second).

For sure duplex Ultrasound it is not „objective“, as it is operator dependent, but this, at least has to be mentioned. And your new method, even if the images are gorgeous, aren’t they also operator dependent as far as the interpretation is “human”? Thus, you cannot simply leave out the most used tool in your list.

[Answer 5]

Thank for your decent suggestion. We revised this table as your suggestion.

[Comment 6]

In this sentence you mix topics: Compression and collateral veins are the domain of deep venous obstruction – and these deep veins NEVER become ablated!

It is in the superficial veins, where we perform ablation (or multiple other treatment options you seem not to be familiar with) – but superficial veins never get incompetent due to Compression …?? It looks like you are coming from deep vein diagnostics and apply the tool to superficial veins, which you seem not to be familiar with. This is a pity, because I am sure, that what you present is of major interest.

[Answer 6]

We revised the Figure 3 again, with standard bar and abbreviations on the figure legend.

[Comment 7]

Then you talk about valvular and non-valvular insufficiency… where might I instruct myself in non-valvular insufficiency of superficial leg veins? (I assume we are talking about superficial veins, as this is the name of your article…?

If this is something new, you want to introduce to the reader, you have to describe it in the background.

[Answer 7]

We revised the Figure 3 again, with standard bar and abbreviations on the figure legend.

[Comment 8]

Then you talk about valvular and non-valvular insufficiency… where might I instruct myself in non-valvular insufficiency of superficial leg veins? (I assume we are talking about superficial veins, as this is the name of your article…?

[Answer 8]

We revised the title as “Discriminating superficial venous valvular insufficiency from non-insufficiency diseases by novel Non-contrast MR with QFlow technique“ for less misleading.

[Comment 9]

I cannot find methods “in the typical location” after background and before Results – I have been reading and re-reading results without understanding anything, just to find methods later on… Methods is always prior to results!

[Answer 9]

The Instructions for authors of Journal of Personalized Medicine are unique in the section arrangement, which the “methods” is “after” the discussion section. (https://www.mdpi.com/journal/jpm/instructions)

To improve the comprehensiveness of this article, we point out our previous literature in “Background “section and further detail the methodology as the “Instructions for author” requested. We will rearrange those Figures numbers as the final suggestion of the reviewers and editors.

[Comment 10]

You write that the patients had a Duplex in supine position prior to MRI – Please explain, why you do this non according to World-Wide Consensus from 2006!! (Coleridge Smith et al) which say, that superficial leg veins have to be explored in standing position?

[Answer 10]

There were some patients receiving Duplex in standard position in this study. However, there are many patients, especially those with leg wounds and knee degeneration, could not tolerate standing examination thus no available data in their standing position.

[Comment 11]

The MRI investigation is performed in recumbent persons too, please describe, how you provoke retrograde flow? If not in any standardized way, the complete measurement lack reproducibility! Venous reflux occurs in standing position and you can apply different methods. Calf compression with standardized cuffs are the only one repeatable with always the same results as to my knowledge

[Answer 11]

Thank for your suggestive comment and point out the major limitation of this study. Ultrasound by experienced operator still the most efficient diagnostic tool in the venous disease of the legs.

Change: Study limitation section

“Third, the TRANCE MR and US were performed at supine position, without details of gravitational and muscular interactions. The retrograde flow is not possibly provoked during TRANCE MR examination and could be better reproduced by US in standing position.”

[Comment 12]

Image 1 a and b

For me it is obvious, which one is the GSV in the legs. Could you please add arrows to the findings – the healthy one on the right leg (both), the healthy one below the onset of reflux in tributaries at knee level left and the reflux in tributaries?

[Answer 12]

We add some marks the figure 1 in this version according to your suggestion.

Change: New figure 1 and changing in the Figure legends.

GSV: normal right side great saphenous vein. Double arrow indicates the diseased great saphenous vein with its tributaries.

[Comment 13]

In table 2 I get lost – suddenly you talk about obstructive and non obstructive – but if you are talking about superficial leg vein reflux discrimination, for the first step you should have excluded those with deep vein obstruction. Of course, I know, that some have both – but if you introduce a new tool (and it looks like a good one), you should first publish the tool in “simple” Legs

[Answer 13]

Thank for your decent comments. We revised this section as your previous suggestion and focus on the superficial venous valvular insufficiency in this article. We are aggressively collecting health individuals by TRANCE MR, for better comparison. However, the healthy volumeters are not easy to recruit.

[Comment 14]

Then, I do not understand the table – you wanted to compare healthy to not healthy, but I only find data of insufficiency, where is the comparison to contralateral?

[Answer 14]

To minimize individual bias in the QFlow analysis, we compared the morbid and nonmorbid limbs of the same patients. All parameters in table 2 used the ratio of morbid limb to normal limb in each venous segment.

Change

Those explanation has been revised in the end of page 3.

[Comment 15]

In the Flow chart you write “Non insufficiency” – but I thought, you include only people with insufficiency at one leg??

[Answer 15]

Yes, one-leg involved patients included. The Flow chart was revised for better comprehensive.

[Comment 16]

In the Flow chart you write “Non insufficiency” – but I thought, you include only people with insufficiency at one leg??

[Answer 16]

Yes, one-leg involved patients included. The Flow chart was revised for better comprehensive.

[Comment 17]

Discussion

You write, Duplex is time consuming (in experienced hands 10 minutes the most to map a patient…?) – but this investigation takes !! one hour!!

[Answer 17]

It may represent the value of experienced operator in US survey. In our institution, the Duplex performed in an ultrasound center, which is not done by the same physician in the clinics. Thus the Ultrasound reports are not quite comprehensive to the patients, their family even the referral vascular surgeons. The TRANCE MR, although time consuming, is not very expensive in our country. (less than 250 USD including QFlow analysis). The MRI operating time is now shorter than 30 min (venous system).

[Comment 18]

IVUS – which has nothing to do with superficial vein disease.

Please sort out all those indications concerning only deep vein disease.

[Answer 18]

We modified the description on the tool introduction and emphasized more on the superficial venous systems

Changing: But the IVUS is invasive, and only provides the detail of venous lumen, without information of superficial venous system.

[Comment 19]

In your Text Conclusion you do not write about consequences, in the Abstract yes

[Answer 19]

We revised the conclusion similar to the abstract.

Thank you for the informative and careful reviewing our manuscript. We learn lot during the revision this article.   The complete response to the reviewers, are attached in the file.

Round 2

Reviewer 2 Report

Reviewer 2
[Comment 1]
Very interesting topic, as it is, indeed, difficult to evaluate venous disease “Objectively”. And TRANCE MRI seems to be an interesting point, the images are gorgeous.

The authors mix superficial and deep venous disease. The whole article should be rewritten separating both.

[Answer 1]
We appreciated your informative comments for this article. We emphasized the components of superficial and deep venous disorders in this disease entitle (venous disease in lower extremity). We defined the group in this version (reflux vs. non-reflux in great saphenous vein), to reduce the misleading.

You compared refluxive versus healthy veins? Or non-reflux pathology – I do not understand which then is this pathology? Superficial vein thrombosis?

[Comment 2]
Abstract:
methods: “Limb disease” please specify: Superficial reflux? Deep obstruction or reflux?
[Answer 2]
We revised the paragraph as your suggestion and defined as superficial reflux. Change: Abstract, Methods, line 2
53 with unilateral superficial venous reflux in the leg.

you will have to change it in the methods of the main text, also

[Comment 3]
Background:
The title of your article refers to “Superficial Veins” – why then do you only measure in GSV (one superficial) but not in SSV, not in tributaries? Instead in three deep veins, one from the pelvic system? Of course, the deep veins are interesting when studying superficial ones. But then you only study one?

[Answer 3]
The major reason is that the GSVs are more constant than SSVs in supine position by 1.5T MRI. We just started to analyze SSV recently but did not gain meaningful conclusion yet. ok

[Comment 4]
You talk about reflux and obstruction very indistinctly, but obstruction is a primary deep vein problem (seldom also superficial), whereas Reflux is THE problem with superficial venous insufficiency.

 [Answer 4]
Thank for your comments. We add description in the re-write
We specified the reflux as “superficial venous reflux” in this version, to reduce misunderstanding of the readers. You have to explain this in the introduction and also in methods and results, mot only in the conclusion. This is the main topic of the article and the correction should be performend throughout the paper.

Changed: Conclusion line 262 to 265

[Comment 5]
When you talk about different ways to measure venous insufficiency and obstruction you cannot leave out Duplex Ultrasound or Air Plethysmography, which are the most used (the first) and the only one exactly repeatable and reliable until now (the second).

For sure duplex Ultrasound it is not “objective“, as it is operator dependent, but this, at least has to be mentioned. And your new method, even if the images are gorgeous, aren’t they also operator dependent as far as the interpretation is “human”? Thus, you cannot simply leave out the most used tool in your list. [Answer 5]

Thank for your decent suggestion. We also brief introduce the air plethysmography in the discussion section.

Duplex Ultrasound should be mentioned in Introduction. Thanks for clarifying, that you meant 18 legs have reflux and 35 legs have no reflux. But if 35 legs out of 53 has no reflux. But you excluded 72 patients because they have no “single leg” Reflux (this means, both legs?) how then, of the remaining 53 legs not ALL studied have reflux? So you can only evaluate 18 legs? Because the topic is superficial leg vein reflux?

[Comment 6]
In this sentence you mix topics: Compression and collateral veins are the domain of deep venous obstruction – and these deep veins NEVER become ablated!
It is in the superficial veins, where we perform ablation (or multiple other treatment options you seem not to be familiar with) – but superficial veins never get incompetent due to Compression ...?? It looks like you are coming from deep vein diagnostics and apply the tool to superficial veins, which you seem not to be familiar with. This is a pity, because I am sure, that what you present is of major interest.
[Answer 6]
The “collateral veins” is better described as tributaries.
Change: Background at bottom: TRANCE-MRI helps to not only exclude the venous compression but also reveal the major tributaries, thus providing better guidance for venous ablation.

[Comment 7]
Then you talk about valvular and non-valvular insufficiency... where might I instruct myself in non-valvular insufficiency of superficial leg veins? (I assume we are talking about superficial veins, as this is the name of your article...?

If this is something new, you want to introduce to the reader, you have to describe it in the background.

[Answer 7]
We revised the Background again, with the technique trick in the vessel acquisition and background suppression of TRANCE MRI. And list some document of TRANCE MRV from our team with a brief introduction. Meanwhile, a new Figure 1 (TRANCE MR, principle for vessels) for better comprehension for the readers, was added in this version.

Change:Background
Patients were suspected as venous origin when they came with tortious varicose veins on calves, asymmetric swollen legs and watery ulcer around the gaiter area of their feet. Venous diseases in those legs may be caused by superficial venous reflux from valvular insufficiency, occlusion of deep venous system by thrombi and compression the pelvic mass. The resulting signs include spider veins, foot pigmentation, claudication, stasis ulcers, swollen limbs with deep vein thrombosis and the fatal consequece of the pulmonary embolism.

All images of the arterial systems are reconstructed by three-dimensional turbo spin-echo (TSE) at systolic and diastolic periods. During systole, arterial blood flows rapidly and the arteries are black. In diastole phase, blood flow in the arteries is slow and the arteries are bright. Subtraction of the two phased scans made up a 3D data set with only arteries (MRA). Another image of the venous systems (MRV) was evaluated by 3D TSE Short tau inversion recovery (STIR) during the systolic period. STIR provides additional background suppression because the fat and bones are also suppressed. Our clinical applications of this technique have innovated the anatomy of the whole venous system in legs since 2017 and further proven its efficiency in different venous scenarios thereafter.

New Figure 1

[Comment 8]
Then you talk about valvular and non-valvular insufficiency... where might I instruct myself in non-valvular insufficiency of superficial leg veins? (I assume we are talking about superficial veins, as this is the name of your article...? [Answer 8]
We revised the title as “Discriminating reflux from non-reflux diseases of superficial veins in legs by novel Non-contrast MR with QFlow technique “ for less misleading.
[
Comment 9]
I cannot find methods “in the typical location” after background and before Results – I have been reading and re-reading results without understanding anything, just to find methods later on... Methods is always prior to results!

[Answer 9]

We revised as your suggestion.

[Comment 10]
You write that the patients had a Duplex in supine position prior to MRI – Please explain, why you do this non according to World-Wide Consensus from 2006!! (Coleridge Smith et al) which say, that superficial leg veins have to be explored in standing position?

[Answer 10]
There were some patients receiving Duplex in standard position in this study. However, there are many patients, especially those with leg wounds and knee degeneration, could not tolerate standing examination thus no available data in their standing position.
Change: We announce this in the study limitations.

[Comment 11]

The MRI investigation is performed in recumbent persons too, please describe, how you provoke retrograde flow? If not in any standardized way, the complete measurement lack reproducibility! Venous reflux occurs in standing position and you can apply different methods. Calf compression with standardized cuffs are the only one repeatable with always the same results as to my knowledge

[Answer 11]
Thank for your suggestive comment and point out the major limitation of this study. Ultrasound by experienced operator still the most efficient diagnostic tool in the venous disease of the legs.
Change: Study limitation section
“Third, the TRANCE MR and US were performed at supine position, without details of gravitational and muscular interactions. The retrograde flow is not possibly provoked during TRANCE MR examination and could be better reproduced by US in standing position.”

If you did not do anything to provoke a retrograde flow, the only diagnostic tool was the ultrasound, and then you measured typical MRI Measurementes without influencing the blood flow, this is interesting, but I did not understand it from the context.

Thus you should perhaps include in the introduction and conclusion, that you found a correlation between the reflux with ultrasound and morphological changes in the vein objectivable in the MRI – which would then really be operator independent tool.

[Comment 12]
Image 1 a and b
For me it is obvious, which one is the GSV in the legs. Could you please add arrows to the findings – the healthy one on the right leg (both), the healthy one below the onset of reflux in tributaries at knee level left and the reflux in tributaries?
[Answer 12]
We add some marks the figure 1 in this version according to your suggestion.

Change: New figure 1 and changing in the Figure legends.
GSV: normal right great saphenous vein. Double arrow indicates the diseased left

great saphenous vein with its tributaries.

[Comment 13]
In table 2 I get lost – suddenly you talk about obstructive and non obstructive – but if you are talking about superficial leg vein reflux discrimination, for the first step you should have excluded those with deep vein obstruction. Of course, I know, that some have both – but if you introduce a new tool (and it looks like a good one), you should first publish the tool in “simple” Legs
[Answer 13]
Thank for your decent comments. We revised this section as your previous suggestion and focus on the reflux in superficial veins in this article. We are aggressively collecting health individuals by TRANCE MR, for better comparison. However, the healthy volumeters are not easy to recruit and is not enough now.

[Comment 14]
Then, I do not understand the table – you wanted to compare healthy to not healthy, but I only find data of insufficiency, where is the comparison to contralateral?
[Answer 14]
To minimize individual bias in the QFlow analysis, we compared the morbid and nonmorbid limbs of the same patients. All parameters in table 2 used the ratio of morbid limb to normal limb in each venous segment.
Change
Those explanation has been revised in the end of page 3.

[Comment 15]
In the Flow chart you write “Non insufficiency” – but I thought, you include only people with insufficiency at one leg??
[Answer 15]
Yes, one-leg involved patients included. The Flow chart was revised for better comprehensive.

Change: please see flow chart
[Comment 16]
In the Flow chart you write “Non insufficiency” – but I thought, you include only people with insufficiency at one leg??
[Answer 16]
Yes, one-leg involved patients included. The Flow chart was revised for better comprehensive.

[Comment 17]
Discussion
You write, Duplex is time consuming (in experienced hands 10 minutes the most to map a patient...?) – but this investigation takes !! one hour!!
[Answer 17]
It may represent the value of experienced operator in US survey. In our institution, the Duplex performed in an ultrasound center, which is not done by the same physician in the clinics. Thus the Ultrasound reports are not quite comprehensive to the patients, their family even the referral vascular surgeons. The TRANCE MR, although time consuming, is not very expensive in our country. (less than 250 USD including QFlow analysis). The MRI operating time is now shorter than 30 min (venous system).

So, perhaps you should comment on local organizing problems (Which is ok, because you are not the only one with these problems).

[Comment 18]
IVUS – which has nothing to do with superficial vein disease.
Please sort out all those indications concerning only deep vein disease.
[Answer 18]
We modified the description on the tool introduction and emphasized more on the superficial venous systems
Changing: But the IVUS is invasive, and only provides the detail of venous lumen, without information of superficial venous system.

 [Comment 19]
In your Text Conclusion you do not write about consequences, in the Abstract yes [Answer 19]
We revised the conclusion similar to the abstract.

Author Response

March 21, 2021

Journal: JPM (journal of personalized medicine)

Manuscript ID: jpm-1129100-r2

Title: Discriminating valvular insufficiency from non-insufficiency venous diseases by novel Non-contrast MR with QFlow technique

(new title: Discriminating reflux from non-reflux diseases of superficial veins in legs by novel Non-contrast MR with QFlow technique)

Dear Editors, Reviewers and Ms.Christa Sas

We are submitting our manuscript entitled “Discriminating reflux from non-reflux diseases of superficial veins in legs by novel Non-contrast MR with QFlow technique " for consideration of “Journal of Personalized Medicine” after revise.

        Thank you very much again for granting the privilege to us to revise the paper. We have specifically responded to the reviewers’ questions and criticisms point-by-point as follows and add them into this version. Any changes in the manuscript can be tracked by the tool of the MS Word and be marked by underline.

Reviewer 2
[Comment 1]
Very interesting topic, as it is, indeed, difficult to evaluate venous disease “Objectively”. And TRANCE MRI seems to be an interesting point, the images are gorgeous.

The authors mix superficial and deep venous disease. The whole article should be rewritten separating both.

[Answer 1]
We appreciated your informative comments for this article. We emphasized the components of superficial and deep venous disorders in this disease entitle (venous disease in lower extremity). We defined the group in this version (reflux vs. non-reflux in great saphenous vein), to reduce the misleading.

You compared refluxive versus healthy veins? Or non-reflux pathology – I do not understand which then is this pathology? Superficial vein thrombosis?

Reply for 2nd Revise

The superficial veins in non-reflux group are mostly in normal size, with competent valve function.

We had add this description in the result section.

The two groups were similar in terms of age, sex, substance use, comorbidities, and co-existence of leg ulcers. QFlow analysis through TRANCE-MRI involved the evaluation of SV (mL), FFV (mL), MF (mL), SD (cm), and MV (cm) in the external iliac veins, femoral veins, popliteal veins, and GSVs in the 53 patients. “The superficial veins in non-reflux group are mostly in normal size, with competent valves function”. To minimize individual bias in the QFlow analysis, we compared the morbid and nonmorbid limbs of the same patients.

[Comment 2]
Abstract:
methods: “Limb disease” please specify: Superficial reflux? Deep obstruction or reflux?
[Answer 2]
We revised the paragraph as your suggestion and defined as superficial reflux. Change: Abstract, Methods, line 2
53 with unilateral superficial venous reflux in the leg.

you will have to change it in the methods of the main text, also
Reply for 2nd revise

We uniformed the term of reflux in this version.

[Comment 4]
You talk about reflux and obstruction very indistinctly, but obstruction is a primary deep vein problem (seldom also superficial), whereas Reflux is THE problem with superficial venous insufficiency.

 [Answer 4]
Thank for your comments. We add description in the re-write
We specified the reflux as “superficial venous reflux” in this version, to reduce misunderstanding of the readers. You have to explain this in the introduction and also in methods and results, mot only in the conclusion. This is the main topic of the article and the correction should be performend throughout the paper.

Reply in 2nd revise

We uniformed the term of reflux in this version.

[Comment 5]
When you talk about different ways to measure venous insufficiency and obstruction you cannot leave out Duplex Ultrasound or Air Plethysmography, which are the most used (the first) and the only one exactly repeatable and reliable until now (the second).

For sure duplex Ultrasound it is not “objective“, as it is operator dependent, but this, at least has to be mentioned. And your new method, even if the images are gorgeous, aren’t they also operator dependent as far as the interpretation is “human”? Thus, you cannot simply leave out the most used tool in your list. [Answer 5]

Thank for your decent suggestion. We also brief introduce the air plethysmography in the discussion section.

Duplex Ultrasound should be mentioned in Introduction.

Reply for 2nd revise

We add short summary in the “introduction” and add two references for the readers.

“Ultrasound examination (US) is the standard procedure for venous disease of the legs and could provide most information by experienced operators.[7,8]”

  1. Coleridge-Smith P, Labropoulos N, Partsch H, Myers K, Nicolaides A, Cavezzi A. Duplex ultrasound investigation of the veins in chronic venous disease of the lower limbs--UIP consensus document. Part I. Basic principles. Eur J Vasc Endovasc Surg 2006;31:83-92.
  2. Marsden G, Perry M, Kelley K, Davies AH, Guideline Development G. Diagnosis and management of varicose veins in the legs: summary of NICE guidance. BMJ 2013;347:f4279.

Thanks for clarifying, that you meant 18 legs have reflux and 35 legs have no reflux. But if 35 legs out of 53 has no reflux. But you excluded 72 patients because they have no “single leg” Reflux (this means, both legs?) how then, of the remaining 53 legs not ALL studied have reflux? So you can only evaluate 18 legs? Because the topic is superficial leg vein reflux?

Reply to 2nd revise.

Yes, the most important reason is to focus on the superifical vein reflux in legs. Only typical patients with one leg disease were included in the 18 patients.

[Comment 11]

The MRI investigation is performed in recumbent persons too, please describe, how you provoke retrograde flow? If not in any standardized way, the complete measurement lack reproducibility! Venous reflux occurs in standing position and you can apply different methods. Calf compression with standardized cuffs are the only one repeatable with always the same results as to my knowledge

[Answer 11]
Thank for your suggestive comment and point out the major limitation of this study. Ultrasound by experienced operator still the most efficient diagnostic tool in the venous disease of the legs.
Change: Study limitation section
“Third, the TRANCE MR and US were performed at supine position, without details of gravitational and muscular interactions. The retrograde flow is not possibly provoked during TRANCE MR examination and could be better reproduced by US in standing position.”

If you did not do anything to provoke a retrograde flow, the only diagnostic tool was the ultrasound, and then you measured typical MRI Measurementes without influencing the blood flow, this is interesting, but I did not understand it from the context.

Thus you should perhaps include in the introduction and conclusion, that you found a correlation between the reflux with ultrasound and morphological changes in the vein objectivable in the MRI – which would then really be operator independent tool.

Reply for 2nd revise

We add a brief summary in the introduction

In this study, we evaluate the possible correlation between reflux with ultrasound and morphologic changing from the TRANCE MRI, and studied the hemodynamic patterns obtained from QFlow analysis through TRANCE-MRI to help differentiate reflux in superficial veins further.

and in “conclusion”

TRANCE-MRI is a potential tool for verifying the characteristic ultrasound features of reflux venous disease in lower extremities

[Comment 17]
Discussion
You write, Duplex is time consuming (in experienced hands 10 minutes the most to map a patient...?) – but this investigation takes !! one hour!!
[Answer 17]
It may represent the value of experienced operator in US survey. In our institution, the Duplex performed in an ultrasound center, which is not done by the same physician in the clinics. Thus the Ultrasound reports are not quite comprehensive to the patients, their family even the referral vascular surgeons. The TRANCE MR, although time consuming, is not very expensive in our country. (less than 250 USD including QFlow analysis). The MRI operating time is now shorter than 30 min (venous system).

So, perhaps you should comment on local organizing problems (Which is ok, because you are not the only one with these problems).

Reply to 2nd revise

We add a brief comment on the discussion

US is operator-dependent and lacking information of pelvic and abdominal areas. “However, many institutions including us, the Duplex performed exclusively in ultrasound centers, which are not done by the same physician in the clinic, which required additional communications between stuffs to gain sufficient information.”  

Sincerely,

Yao-Kuang Huang, MD, PhD

Division of Thoracic and Cardiovascular Surgery

Chia-Yi Chang Gung Memorial Hospital, Putz, Taiwan.

Fax: 886-975368209

E-mail: huang137@icould.com

This manuscript is a resubmission of an earlier submission. The following is a list of the peer review reports and author responses from that submission.

Round 1

Reviewer 1 Report

QFlow as an indicator to discriminate varicose and nonvaricose venous diseases through triggered angiography non–contrast-enhanced sequence magnetic resonance imaging

Thank you very much for asking me to review this fascinating paper with very impressive images, particularly in the attached MP4 video.

Unfortunately, although the images are really quite beautiful and are fascinating for a venous Surgeon to see, it would appear that the authors have not taken any advice from a venous Surgeon as to what would be regarded as useful or even unacceptable methodology.

They are distinguishing between "non-varicose" and "varicose" disease, whereas venous surgeons are looking for reflux primarily, and then stasis and obstruction.

Reflux can either be passive (diastolic) or gravitational, or active (systolic) due to muscular action.

In the method section, the authors point out that the patients have undergone duplex examination (they call colour Doppler) in the supine position. Absolutely no venous Surgeon who understands anything to do with venous disease would ever perform a reflux examination for varicose veins in the supine position as the most common form of venous reflux, gravitational or passive (diastolic) reflux is absent when supine.

Therefore the authors do not have a proper comparator to assess their MRI findings against. It would be hard to see how MRI, performed in the supine position, could check for gravitational (passive) reflux but if they had managed to do this, they would need to compare it against a proper colour flow duplex ultrasound examination, performed in the erect position by a vascular technologist using standardised protocols.

As they have no acceptable comparator of disease or no disease, then all of their MRI images and measurements, although beautiful, cannot be assessed against anything regarded as pathological by venous surgeons.

Reviewer 2 Report

interesting, well planned publication. Perhaps it was necessary to perform comparative Doppler ultrasound tests. I suggested that they be blinded